# Genital Lymphedema after Cancer Treatment: A Narrative Review

**DOI:** 10.3390/cancers14235809

**Published:** 2022-11-25

**Authors:** Stéphane Vignes

**Affiliations:** Department of Lymphology, Centre National de Référence des Maladies Vasculaires Rares (Lymphoedèmes Primaires), Hôpital Cognacq-Jay, 15, rue Eugène-Millon, 75015 Paris, France; stephane.vignes@cognacq-jay.fr; Tel.: +33-(0)1-45-30-81-00; Fax: +33-(0)1-45-30-81-34

**Keywords:** lymphedema, genital, treatment, surgery

## Abstract

**Simple Summary:**

Genital lymphedema may develop in males and females after cancer treatment. It is frequently associated with lower limb lymphedema, and is responsible for discomfort, cosmetic disfigurement and functional disturbances. Impacts on body image, sexual function and quality of life are major and difficult to explore, because cancer treatment itself and lymphedema are tightly intricated. Local complications and cellulitis may occur. Usual lymphedema therapies, like bandaging and elastic compression, are poorly adapted to these sites. Surgery, essentially based on cutaneous resection techniques, is the main symptomatic treatment; it has good efficacy, in adults and children, with possible recurrence requiring reintervention.

**Abstract:**

Genital lymphedema may affect males and females after cancer treatment (gynecological, such as cervical, uterine or ovarian, melanoma, prostate, anus…). It is frequently associated with lower limb lymphedema, and is responsible for discomfort, cosmetic disfigurement and functional disturbances. Impacts on body image, sexual function and quality of life are major, and difficult to explore because cancer treatment itself and lymphedema are so closely interwoven. Local complications, e.g., papillomatosis, warty growth, lymph vesicles with embarrassing lymph oozing and cellulitis, may occur. Usual lymphedema therapies, like bandaging and elastic compression, are poorly adapted to these sites. Surgery, essentially based on cutaneous resection techniques, is the primary symptomatic treatment; it achieves good efficacy, in adults and children, with possible recurrence requiring reintervention.

## 1. Introduction

Lymphedema is initially related to an accumulation of lymph that increases tissue volume in association with structural modifications, especially fibrosis with skin thickening and adipose deposition. These changes lead to a predominantly irreversible state in which the lymphatic component is lower than that of the tissue (fat, skin) [1]. Genital lymphedema is defined as increased volume of the external genitalia: penis, foreskin and scrotum in males, and labia minora and majora in females. Lymphedema may also involve adjacent regions: pubis (mons veneris), groin and inner thigh(s). Lower limb lymphedema is frequently associated with genital lymphedema [2].

Lymphedema is divided into primary and secondary forms. Primary lymphedema is a congenital malformation of the lymphatic network, including lymphatic channels. It is essentially sporadic, sometimes familial (due to *VEGFR3* mutation, called Milroy’s disease, with lower limb lymphedema, present at birth or before 1 year, sometimes with genital involvement [3]). Primary lymphedema may be isolated or syndromic, i.e., representing a clinical sign of a more complex syndrome, often genetic, or with chromosomal abnormality, such as Turner’s syndrome. Some genetic mutations are mostly implicated in primary lymphedema involving genitals [4,5]. In developing countries, filariasis is the main cause of secondary forms of genital lymphedema [5,6,7]. In industrialized countries, although the main cause of secondary lymphedema is cancer treatment [8,9], other possible causes are given in Table 1 [10,11,12,13,14,15,16,17,18,19,20,21,22,23,24,25,26,27,28,29,30]. Herein, we describe genital lymphedema after cancer treatment in males and females, highlighting potential diagnostic pitfalls.

## 2. Frequency and Risk Factors

Because it is not easy to obtain clear results when looking for genital lymphedema in a database, like PubMed, especially for females, we used a variety of search terms to describe genital lymphedema, e.g., lymphangioma or circumscriptum lymphangioma, thus probably underestimating its identification and frequency. In most articles, secondary lymphedema after cancer treatment mostly concerned the lower limb, with very few details on genital involvement. 

Secondary genital lymphedemas may arise after various treatments for a wide variety of cancers of the pelvic region and lower limb (e.g., uterine, cervical, ovarian, prostate, rectum, melanomas, Hodgkin or non-Hodgkin lymphomas), including surgery with more-or-less extensive lymph-node excision (inguinal, iliac, lumboaortic, pelvic), brachytherapy and/or external radiotherapy [8,9]. Secondary lower limb lymphedema after cancer treatment is little-known because its definition is not consensual, with a large frequency range for a given cancer and its treatment, which often differs according to cancer classification, and it is rarely mentioned [31,32]. These difficulties are related to bilateral lymphedema involvement, hence lacking a control limb, and to its frequent assessment with subjective and not objective criteria, especially without preoperative limb measurements. According to a retrospective study on 356 women [32], 15% of them developed lower limb lymphedema after gynecological cancer treatment, with median onset at 240 (range: 3–1415) days. Furthermore, 38.9% of those 54 patients developed lower limb lymphedema within 6 months and 85.2% within 2 years. In other series, 15–48% of women treated for gynecological cancer developed lower limb lymphedema, as opposed to 30% after melanoma or prostate cancer [9,31]. Details of genital involvement are rarely mentioned, without specific questioning, and exploring intimacy is difficult. It is important to be aware that genital lymphedema is poorly known by healthcare professionals, with very little or no specific training in its diagnosis, treatment and management. Therefore, online educational resources—printed offline material and videos for health professionals and patients, with collaborative and multidisciplinary decisions—are sorely needed [33]. For patients with lower limb lymphedema, genital involvement frequency is either impossible to determine or unclear because of the lack of specific questions or because it appears more-or-less a long time after limb lymphedema onset and follow-up had been too short. The median time to consulting a specialist from the first onset of genital lymphedema after cancer treatment is 1.4 years [34].

Risk factors for lower limb lymphedema are very poorly known and vary according the different studies. For cervical cancer patients, the number of removed lymph nodes and radiotherapy (external with or without brachytherapy) are risk factors, while obesity and the numbers of removed lymph nodes are risk factors for uterine cancer patients. For ovarian cancer, risk factors have not yet been clearly identified [35]. In other studies, body mass index (BMI) ≥ 25 kg/m^2^, radiation therapy, cancer stage, high numbers of excised lymph nodes and lymphocyst formation were risk factors [31,32,36]. Moreover, risk factors and frequency for genital lymphedema were not specifically investigated.

## 3. Physical Examination

Genital lymphedema is almost always associated with unilateral or bilateral lower limb lymphedema and very rarely isolated [2,34]. Unlike limb lymphedema, no consensual technique exists to appreciate genital volume but a patient’s symptoms and signs may be quantified with a specific scale ranging from 0 to 9: the Genital Lymphedema Score (GLS). GLS is comprised of six items: sensation of heaviness, tension, swelling, lymphedema-induced urinary disorders, cutaneous lymphatic cyst and genital lymphorrhea [37]. For lower limb lymphedema, the presence of pathognomonic Stemmer’s sign confirms lymphedema: increased skin thickness, or the inability to pinch the skin of the dorsal face or base of the second toe [38]. Nevertheless, proximal (involving only one or both thighs) secondary lymphedema without Stemmer’s sign is frequently observed. Cancer history and its treatment, and the patient’s sensations then allow confirmation of the lymphedema diagnosis [35].

### 3.1. In Males

Lymphedema may involve the penis, foreskin and/or scrotum to different extents; sometimes only the penis or the scrotum, or more frequently both, with possible changes during its evolution. Initially, lymphedema may be fluctuant and reversible after decubitus positioning, but after several weeks to months becomes persistent. Lymphedema volume varies from moderate to severe, with possible discomfort walking, rubbing against the thighs (enhanced when lymphedema involves the lower limb(s)), and wearing clothing, and might be noticeable by others. Genital lymphedema volume might lead to descent of the scrotum with the sensation of heaviness and/or pain, and may have a negative impact on some activities, like biking (Figure 1). The lymphedematous penis may become abnormally shaped, assuming a characteristic saxophone form (Figure 2). Foreskin involvement may lead to urine entrapment within the pseudo-foreskin cavity and its release in multiple “spurts”. Increased scrotum volume is associated with skin thickening (similar to Stemmer’s sign). When the scrotum volume is massive, the penis may be buried and appear smaller or is even invisible. “Buried” refers to the penile shaft engulfed within the prepubic skin, with a partially or totally hidden penis, also called trapped or concealed [26,39]. Local cutaneous abnormalities may coexist with genital lymphedema: lymphatic vesicles with possible lymph oozing (representing a portal of entry for bacteria that can increase susceptibility to cellulitis), papillomatosis, hyperkeratosis and/or warty growth(s), especially on the scrotum. 

### 3.2. In Females

Genital lymphedema, sometimes called acquired vulvar lymphedema circumscriptum, corresponds to increased volume of labia minora and/or majora, sometimes unilateral or asymmetric. This larger volume causes spontaneous discomfort, rubbing, frequently pruritus and lymphatic vesicles (Figure 3), and can spread to the internal surfaces of thighs, with possible lymph oozing. Cutaneous hypertrophy of the labia and mons veneris is sometimes very voluminous, giving the appearance of papillomatosis, and/or warty growths, which may be mixed with genital warts or condylomas or, less frequently, with malignant lesions [40]. In very rare cases, genital lymphedema appears as multiple polypoid, verrucous nodules involving the vulva that frighten the patient, inciting her to consult. As for males, obesity seems to be a risk factor. Biopsies are not necessary if the cancer history is known and the diagnosis is clinically evident. When obtained, histology of those specimens found dilated lymphatic channels (lymphangiectasia) in the papillary and reticular dermis, fibrosis, dermal edema, and epidermal changes (papillomatous and hyperkeratotic epidermis) (Figure 4), without any sign of malignancy or granulomas [41]. 

## 4. Complications

Most complications of genital lymphedema are similar in males and females, and others seem to be more specific to each sex. Males with foreskin involvement or a buried penis may have difficulty urinating in a continuous stream. Sexual activity may be affected but is difficult to investigate, because of the frequent association with a causal pathology (such as prostate cancer). For females, frequent itching and lymph oozing represent major sources of discomfort in everyday life. Papillomatosis and increased volume of the labia cause notable disability, have a negative cosmetic appearance, cause discomfort, and carry the risk of breakage and lymph oozing, hence also affecting her libido and sexual activity. Intercourse may increase lymph oozing, contributing to diminished libido and sexual activity or complete cessation.

Cellulitis, dermo-hypodermitis due to β-hemolytic streptococci, is the most frequent lymphedema complication; lymphedema alone increases the cellulitis risk by 70 [42]. This bacterial infection is characterized by the sudden onset of high fever and chills, malaise, sometimes headaches, and then (sometimes several hours later) redness, warmth, pain and increased volume of the lymphedematous region. Cellulitis may begin in a lower limb and spread to the genitalia or affect only the genitals. In a series of 34 women with genital lymphedema after cancer treatment, 68% reported one or more previous cellulitis episodes affecting the lower limb and sometimes genitalia [34]. Patients report intense pain when cellulitis spreads to the genitals, rendering them unable to walk. 

Bacterial entry portals include toe-web intertrigo, but also folliculitis of the pubis or, more frequently, oozing lymphatic vesicles. Adapted antibiotic therapy (e.g., amoxicillin) is required to treat cellulitis and fever disappears within 2–3 days and redness in 1 week, while volume may return to its previous state after several weeks. Cellulitis may recur frequently and then require long-term antibiotic prophylaxis, based on benzathine–benzylpenicillin G for a prolonged—but still undefined—duration. Treatment of the entry site is also required: lymphatic vesicles, toe-web intertrigo, folliculitis…

## 5. Quality of Life

Quality of life has been more thoroughly evaluated for females because of their more numerous post-cancer treatment events than for males. It is markedly altered for women after cancer treatment, especially gynecological cancers, including ovarian, cervical and uterine. Pelvic region symptoms after those treatments include the bowels, urinary tract, lymphatic system and genitals. Currently available quality-of-life scales do not specifically address genital lymphedema, but lymphedema, especially of the lower limb, is included in some questionnaires. 

External radiotherapy and/or brachytherapy frequently are used to treat cervical cancer and frequently have complications, such as bladder symptoms (urinary urgency, incontinence, straining to initiate voiding, bladder-emptying problems, night-time micturition), genital problems (dyspareunia, vaginal dryness, tight vagina). The direct impact of lymphedema is difficult to appreciate but genital volume, papillomatosis or warty lesion appearance, and particularly vesicles with lymph oozing dramatically diminish body image and quality of life. Hence, lymphedema’s negative impact on quality of life may increase anxiety, preexisting mental disorder, depression and social difficulties, sometimes associated with secondary treatment-induced menopause [43]. 

For men, body image, masculinity and quality of life are deteriorated by the treatment of genital cancer, like prostate or penis, including erectile dysfunction [44]. Impact of the cancer treatment (e.g., prostate, with possible impotence, urinary incontinence) is combined with that of the lymphedema itself [45]. Penile deformation, increased volume of the genitals, potential buried penis and involvement of the pubic area in sexual activities and/or global discomfort affect the quality of life.

## 6. Explorations

### 6.1. Diagnosis

The diagnosis is usually made easily based on clinical findings according to the patient’s cancer history. Unlike for primary lymphedema, no specific exploration, e.g., lymphoscintigraphy, is required. Lower limb lymphoscintigraphy, with images obtained 40 min after injecting 99m technetium-labeled colloidal albumin subcutaneously into the first web space of both feet may show the radiotracer pathway the scrotum [46]. Scrotum lymphedema is frequently associated with hydrocele (40%), which should be confirmed by ultrasonography [2]. Sometimes, a large hydrocele may be mistakenly considered lymphedema. Pertinently, hydrocele treatment is specific and differs from that of lymphedema but may be done during the same surgical intervention associating plastic and urological surgeries. Ultrasonography is able to confirm the increased thickness of the skin and thus lymphedema. Early diagnosis may be possible only with site-specific lymphography such as magnetic resonance, indocyanine green or nodal lymphography. It is a key to an ideal management of genital lymphedema to prevent complications seen in advanced genital lymphedema [47,48,49].

### 6.2. Pitfalls to Avoid

Two types of stumbling blocks may lead to erroneous diagnosis and/or treatment of genital lymphedema in a cancer setting. The first is the onset of genital lymphedema in the absence of a known cancer. At this time, before leaning towards the diagnosis of primary lymphedema, a compressive lesion should be sought with computed-tomography (CT) scan and/or magnetic resonance imaging (MRI) and, if necessary, positron-emission–tomography (PET) scan. Abdominal or pelvic compression is very rarely caused by a benign lesion. Importantly, genital lymphedema may reveal and precede the diagnosis of cancer (colon, bladder, rectum…). Colonoscopy with biopsies may rule out colon cancer and asymptomatic chronic inflammatory bowel disease, like Crohn’s disease. In particular, the potential association of genital lymphedema revealing gastric linitis plastica should systematically be sought with gastric endoscopy, even in the absence of clinical signs [50].

The second potential pitfall that may lead to diagnosis of secondary lymphedema after cancer treatment, entails ruling out a cancer relapse. Secondary lower limb lymphedema occurred after cancer therapy and genital lymphedema appeared later, or genital lymphedema is the only lymphedematous manifestation. Genital lymphedema onset after a more-or-less long interval following cancer treatment should evoke recurrent malignancy. Clinical signs suggesting cancer relapse include the sudden onset of genital lymphedema, pain, localized inflammatory signs and rapidly worsening lymphedema. Complementary explorations are then required, such as abdomen and pelvic CT scan or MRI, and perhaps even a PET scan. Determinations of specific blood markers, such as embryonic carcinoma antigen, squamous cell carcinoma antigen, prostate-specific antigen, can be useful.

## 7. Treatments

### 7.1. Non-Surgical Therapies

For lower limb lymphedema, complete (or complex) decongestive therapy is the mainstay and first management choice [51]. It includes low-stretch bandage, manual lymph drainage, meticulous skin care and exercises [52]. Obviously, compressive therapy (bandage, wrap) is more difficult than for limb lymphedema. Because of the shape and location of genital lymphedema, bandaging and a fitted elastic compression garment can be difficult to apply and is often impractical. Low-stretch bandages using bands with less than 100% elasticity may achieve volume decrease, especially for lymphedema of the penis during the intensive phase of treatment (Figure 5) [53]. Bandaging may also be useful to prepare for surgical resection, to reduce the volume and accumulated static lymph, and to soften the skin before surgery [51]. Patient education is essential for daily genital washing, application of moisturizing creams to prevent skin breaks and the risk of cellulitis. Patients and sometimes a family member are taught how to self-bandage at home. Many patients use over-the-counter or custom-made expansive, elastic shorts or panties. Tolerance is frequently poor and proximal compression enhances the risk of increasing limb lymphedema, which then requires the patient to also wear thigh-high compression stockings. Specific tools have been developed to treat genital lymphedema, such as pads (ex: Chip Pad Genifit^®^ Sigvaris; Figure 6) or compressive shorts with removable pad (Mobiderm Intimate Short^®^ Thuasne). For females, compression is more difficult to adapt and compression therapy of the vulva is inadequate or poorly tolerated.

Intermittent pneumatic compression for lower limb lymphedema is not always recommended or only very cautiously for these patients, because it might induce or aggravate genital lymphedema. It consists of high pressure applied with intermittent pneumatic compression on the lymphedematous leg and may lead to congestion of the root of the limb, thereby increasing lymph flow into the soft tissues of the genitals [54]. The recent compression devices use lower pressure and include pelvic compression. Patients with genital lymphedema are often taught manual lymph drainage and self-drainage, but no clear evidence of its impact on volume reduction has been reported [55]. Moreover, physiotherapists can be reluctant to do manual lymph drainage and bandaging for genital lymphedema related to embarrassment, taboo and risk of accusations of sexual abuse. Exercises are a major component of complex decongestive therapy to treat limb lymphedema but have not been clearly evaluated for genital lymphedema.

### 7.2. Surgery

Surgical objectives are multiple: reduce lymphedema volume (cutaneous resection, liposuction); restore “physiological” lymphatic function (lymphovenous anastomosis, lymph-node transfer); and/or limit or suppress lymphatic complications, such as vesicles with lymph oozing and cellulitis. Published surgical recommendations for lymphedema management contain no mention of genital lymphedema and have addressed only upper and lower limbs [56].

For the male patient, the objectives are numerous: restore an acceptable cosmetic appearance close to normal, debulk excessive lymphedematous tissue that notably increases the weight of the scrotum and induces its descent (pendulous scrotum) and, finally, preserve urinary and sexual functions [57]. Various procedures have been proposed for male adults and children: only surgical debulking to remove lymphedematous tissue by scrotoplasty, suction-assisted lipectomy, surgical lipectomy of the suprapubic region or penis plasty, lymphedematous tissue resection and reconstruction of soft tissue and lymphatic structure using superficial circumflex iliac artery perforator lymphatic flap transfer [58]. Reconstructive procedures include ultramicrosurgical lymphatico-venular anastomosis [59,60]. Several techniques may be combined for the same patient [51,61]. Circumcision alone may be proposed for lymphedema essentially localized to the foreskin [62]. After cutaneous resection, reconstruction of skin defects may be achieved with local skin flaps for boys or, more specifically, split-thickness skin grafts, surgical debulking of the scrotum or penile reconstruction with adjacent skin graft is an option [39,63]. These surgeries have few adverse effects, estimated at 6%, such as hematoma and wound dehiscence [51]. Concomitant hydrocele repair or aspiration for adult men and boys may be required. Prophylactic orchidopexy to prevent testicular torsion is another option [64].

For female patients, therapeutic objectives are to improve the cosmetic appearance, decrease lymphedema volume and remove papillomatosis and/or lymph vesicles responsible for embarrassing lymph oozing, spontaneously and/or during sexual intercourse. Lymphedematous tissue resection represents the main surgical intervention, with labia minora and/or majora resection sparing the clitoris [65]. Only half of the women with genital lymphedema underwent surgery, perhaps due to its intimate location or fear of the potential loss of vaginal or clitoral sensitivity.

To date, for genital lymphedema patients in general, “physiological” surgical procedures, such as lymphaticovenous anastomoses, have been attempted but require further assessment before they can be recommended alone or associated with resection surgery [66,67]. When surgeons are experienced and trained to perform these specific techniques, surgery is very useful for genital lymphedema and has very few complications. However, in all cases, surgery including excisional procedures does not treat the underlying pathology and postoperative compression is still required after resection to prevent de novo lymph accumulation and genital volume increase. Repeated surgical procedures may be necessary over the patient’s lifetime due to recurrence. Pertinently, recurrence is frequent, with the reappearance of vesicles with lymph oozing after variable intervals ranging from months to years [51,65].

### 7.3. Other Treatments

For patients with uncomfortable or even debilitating lymphatic vesicles on the vulva or scrotum not requiring skin resection, various non-surgical treatments have been attempted, such as electrocauterization, cryosurgery, argon-laser surgery, sclerosing agent (picibanil) and CO_2_-laser ablation [65,68,69,70].

## 8. Conclusions

After various cancer treatments, males and females may develop genital lymphedema, which is responsible for discomfort, cosmetic disfigurement and functional disturbances. Impacts on body image, sexual function and quality of life are major, and difficult to explore because cancer treatment itself and lymphedema are interwoven. Local complications, especially lymph vesicles with recurrent lymph oozing, are debilitating. Lymphedema therapies (bandaging, elastic compression) are poorly adapted to these sites. Surgery, essentially based on cutaneous resection techniques, is the major symptomatic treatment and has good efficacy, in adults and children, with possible recurrence requiring reintervention. Genital lymphedema is poorly known by healthcare professionals, and health professional training and collaboration with patients is needed to improve management of these very embarrassing disorders.

## Figures and Tables

**Figure 1 cancers-14-05809-f001:**
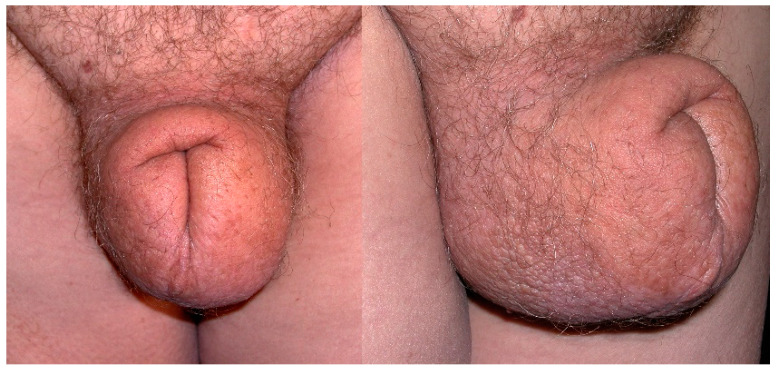
Secondary genital lymphedema in a 58-year-old man after rectal cancer treatment.

**Figure 2 cancers-14-05809-f002:**
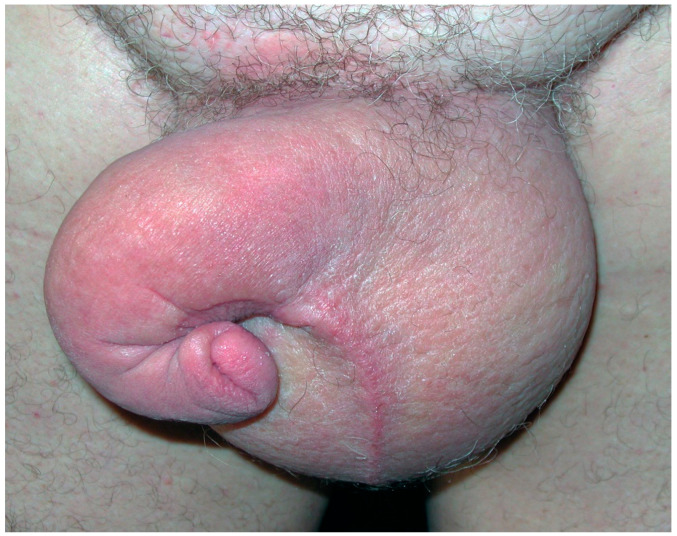
Secondary genital lymphedema in a 65-year-old man after prostate cancer treatment with “saxophone” penis shape.

**Figure 3 cancers-14-05809-f003:**
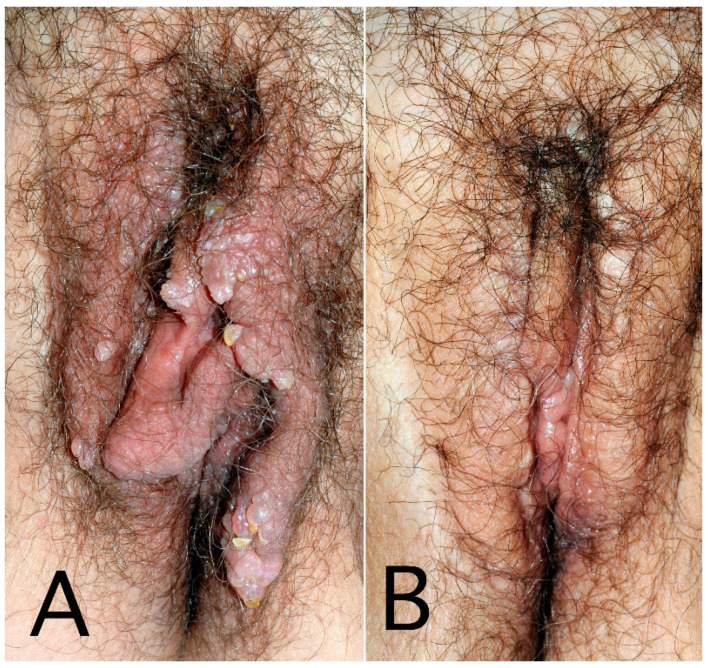
Secondary genital lymphedema with vesicles and warty growth in a 60-year-old woman after cervical cancer treatment: before (**A**) and after 3 resection surgeries (**B**).

**Figure 4 cancers-14-05809-f004:**
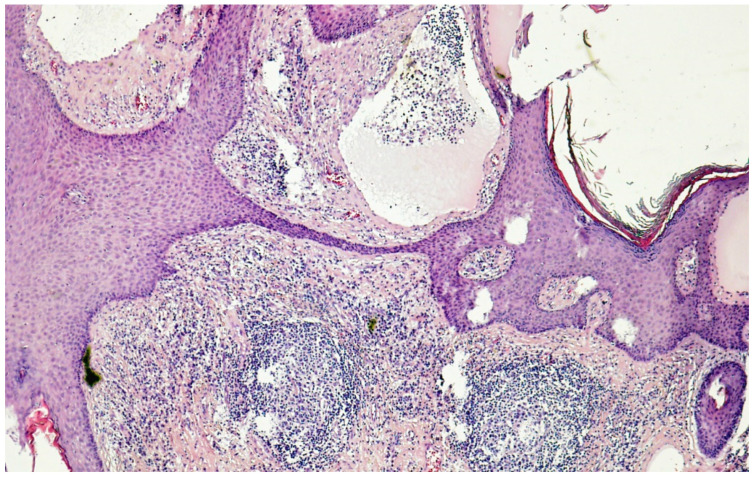
Histology (hematoxylin–eosin–saffron ×100) of genital lymphedema in a 62-year-old woman after cervical cancer treatment: note the hyperkeratotic epidermidis and markedly dilated lymph vessels (lymphangiectasia) predominantly in the papillary dermis.

**Figure 5 cancers-14-05809-f005:**
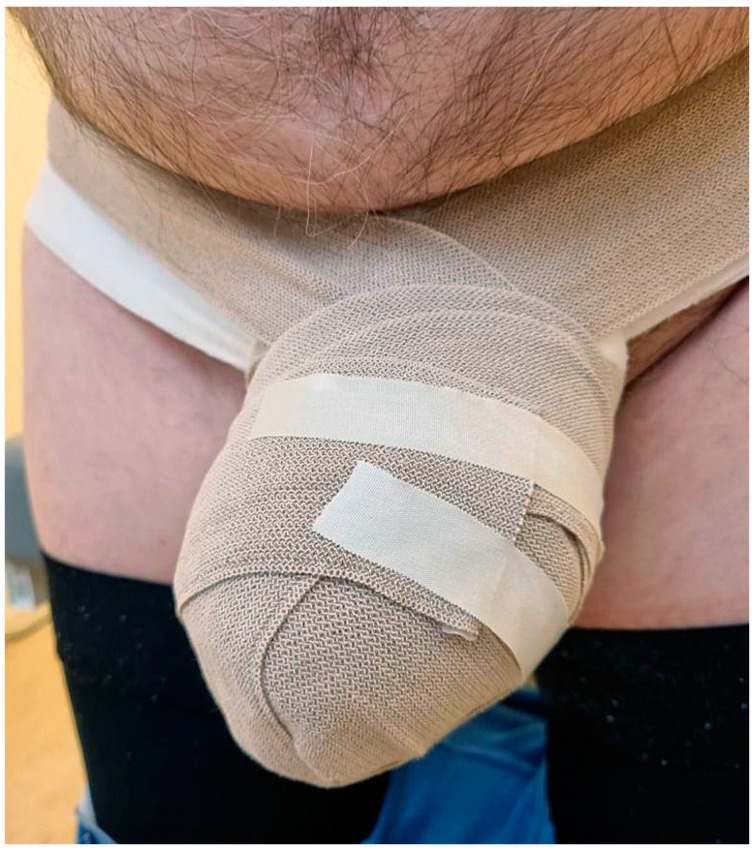
Low-stretch bandages worn by a man with secondary genital lymphedema after rectal cancer treatment.

**Figure 6 cancers-14-05809-f006:**
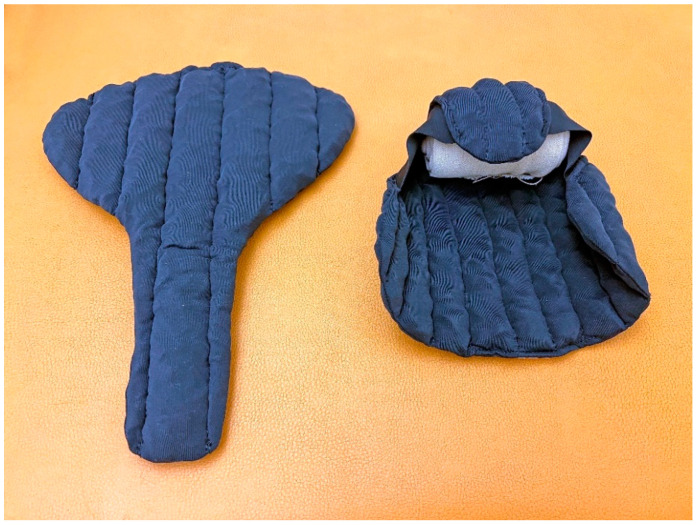
Chip Pad Genifit^®^ garments to treat genital lymphedema in females (**left**) and males (**right**).

**Table 1 cancers-14-05809-t001:** Non-cancer causes of lower limb, and sometimes genital, lymphedema.

Non-Cancer Causes	Diseases
Syndromic primary lymphedema with genetic abnormality [4]	RASopathies (rat sarcoma oncogene protein-associated pathologies: Noonan’s syndrome, *PTPN11, SOS1, RIT1* gene, ORPHA:568062) Emberger syndrome (*GATA2* gene, ORPHA 3226) Lymphedema-distichiasis syndrome (*FOXC2*, ORPHA:33001) Generalized lymphatic dysplasia (*PIEZO1* gene, ORPHA:568062) Hennekam syndrome (*CCBE1* gene, ORPHA:2136)
Primary intestinal lymphangiectasia (Waldmann’s disease, ORPHA:90362) [10]	Protein-losing enteropathy
Chronic inflammatory diseases	Inflammatory bowel diseases: Crohn’s disease (ano-genital granulomatosis), ulcerative colitis [11,12] Hidradenitis suppurativa [13] Chronic rheumatism: ankylosing spondylitis, rheumatoid arthritis, psoriatic arthritis [14,15,16,17]
Infectious diseases	Filariasis (*Wuchereria bancrofti*, *Brugia malayi*, *B. timori*) [7] Kaposi’s disease (endemic, epidemic, acquired immunodeficiency disease syndrome, post-transplantation) [18] *Mycobacterium*, *Chlamydiae trachomatis*, syphilis, actinomycosis, donovanosis, tuberculosis
Retroperitoneal fibrosis	Idiopathic Secondary: cancer, drugs, amyloidosis [19,20]
mTOR (mechanistic target of rapamycin) inhibitors (sirolimus, everolimus)	Post-transplantation [21]
Sexual behavior	Pathomimia, striction [22] Intensive masturbation [23] Local injection of silicone [24]
Morbid obesity, i.e., body mass index > 40 kg/m^2^ [25,26]	Contributes to buried penis
Podoconiosis [27]	Due to long-term exposure to red clay soils
Others	Sarcoidosis [28] Femoropopliteal bypass surgery [29] Peritoneal dialysis [30]

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
