# Peer review of "Genital Lymphedema after Cancer Treatment: A Narrative Review"

_cancers, 2022, doi:10.3390/cancers14235809_

Round 1

Reviewer 1 Report

This is a narrative review on the challenging disease of genital lymphedema. The manuscript describes general information of genital lymphedema, including etiology, incidence, risk factors, manifestations, evaluation, and treatments. Although well described in overall, the manuscript lacks some important points which should be mentioned in genital lymphedema management. The following points should be revised before considering publication.

Title

1. To clarify the article type, please add the term “:a Narrative Review” at the end of the Title.

Diagnosis

2. One of the most challenging points in genital lymphedema management is diagnosis. Since the diagnosis is usually based on physical examination which can be detected only after significantly advanced, genital lymphedema is hardly diagnosed in early stage, making it difficult to cure or control the disease. In most cases, a patient and medical staffs do not notice early changes due to genital lymphedema and usually consider pubic edema an “obese” or physiological changes after pelvic cancer treatments. Early diagnosis is possible only with site-specific lymphography such as magnetic resonance, near-infrared, other lymphography. Early treatment securely prevents sequelae seen in advanced genital lymphedema, and early diagnosis is a key to an ideal management of genital lymphedema. This point should be emphasized with the following references; PMID 27227943, PMID 26992762, PMID 24613771, PMID 22864605, PMID 19914101.

Treatment

3. There are relatively limited numbers of reports of surgical treatments for genital lymphedema, and all previously reported methods should be described in the manuscript. The followings should be described; PMID 32158873, PMID 34756554, PMID 31964116, PMID 21129845, PMID 21093398

Author Response

Dear Editor,

I thank you the reviewer for his constructive comments. I modified the article as follows (in yellow in the text)

  1. To clarify the article type, please add the term “a Narrative Review” at the end of the Title: OK
  2. Diagnosis. The phrase of the reviewer was added with 3 new references. 

    "Early diagnosis may be possible only with site-specific lymphography such as magnetic resonance, indocyanine green or nodal lymphography. It is a key to an ideal management of genital lymphedema to prevent complications seen in advanced genital lymphedema [47–49]".

3.  There are relatively limited numbers of reports of surgical treatments for genital lymphedema, and all previously reported methods should be described in the manuscript. 3 new references were included with 2 phrases in the text to complete surgical techniques.

Reviewer 2 Report

This is a clear and concise summary of current knowledge and practice regarding genital oedema. Good work.

I have attached a few suggestions which I think will enhance the clarity for the reader in a few places. 

Line 40: I think not all lower limb lymphoedema at birth or before 1 year will be called Milroy’s disease, as per the St George’s algorithm. Consider rephrasing? 

Line 78: Reference 33 is correct but are just UK findings, it would give a stronger case to exchange this reference for the international results by the same research group. The recommendations were the same from the larger numbers. This would be:

Noble-Jones, R., Thomas, M.J., Quéré, I., Morgan, K., Moffatt, C. (2021) An international investigation of the education needs of health professionals conservatively managing genital lymphoedema: survey findings. In: Genital Oedema.  Journal of Wound Care (special edition Nov. 2021) pp17-26

Line 203: Is there a reference to go with the hydrocele being associated in 40% of scrotal oedema?

Line 258/259: Reference 51 is a very old study and the machines have changed a great deal since then. The pressures used are usually less and there are advanced pneumatic compression devices with pelvic compression sections included.

Table 1: It would add specificity to add anogenital granulomatosis to the Crohn’s disease/ Chronic Inflammatory diseases row of the table.  For example:

Gordon KD, Brice G, Walker Y, Pollok R, Mortimer P, Slater C. Genital lymphoedema due to anogenital granulomatosis. International Journal of STD & AIDS. 2013;24(2):149-151. doi:10.1258/ijsa.2012.012183 

Author Response

Dear Editor,

I thank you the reviewer for his constructive comments. The text was modified as follows (in yellow in the tetxt):

Line 40: I think not all lower limb lymphoedema at birth or before 1 year will be called Milroy’s disease, as per the St George’s algorithm. Consider rephrasing? 

The new phrase is : ".... sometimes familial (due to VEGFR3 mutation, called Milroy’s disease, with lower limb lymphedema, present at birth or before 1 year, sometimes with genital involvement [3])".

Line 78: Reference 33 is correct but are just UK findings, it would give a stronger case to exchange this reference for the international results by the same research group. The recommendations were the same from the larger numbers. This would be:

OK. The reference was replaced by

Noble-Jones, R., Thomas, M.J., Quéré, I., Morgan, K., Moffatt, C. (2021) An international investigation of the education needs of health professionals conservatively managing genital lymphoedema: survey findings. In: Genital Oedema.  Journal of Wound Care (special edition Nov. 2021) pp17-26

Line 203: Is there a reference to go with the hydrocele being associated in 40% of scrotal oedema?

It's reference no2 (corrected)

Line 258/259: Reference 51 is a very old study and the machines have changed a great deal since then. The pressures used are usually less and there are advanced pneumatic compression devices with pelvic compression sections included.

A new phrase was added. "The recent intermittent compression devices use lower pressure and include pelvic compression."

Table 1: It would add specificity to add anogenital granulomatosis to the Crohn’s disease/ Chronic Inflammatory diseases row of the table.  For example:

Gordon KD, Brice G, Walker Y, Pollok R, Mortimer P, Slater C. Genital lymphoedema due to anogenital granulomatosis. International Journal of STD & AIDS. 2013;24(2):149-151. doi:10.1258/ijsa.2012.012183 

The reference 12 is the same. In the Table, anogenital granulomatosis was added.